# Training Diffusion-based Generative Models with Limited Data

**Zhaoyu Zhang**[1]   **Yang Hua**[1]   **Guanxiong Sun**[2]   **Hui Wang**[1]   **Seán McLoone**[1]

## Abstract

Diffusion-based generative models (diffusion models) often require a large amount of data to train a score-based model that learns the score function of the data distribution through denoising score matching. However, collecting and cleaning such data can be expensive, time-consuming, and even infeasible. In this paper, we present a novel theoretical insight for diffusion models that two factors, i.e., the denoiser function hypothesis space and the number of training samples, can affect the denoising score matching error of all training samples. Based on this theoretical insight, it is evident that minimizing the total denoising score matching error is challenging within the denoiser function hypothesis space in existing methods, when training diffusion models with limited data. To address this, we propose a new diffusion model called Limited Data Diffusion (LD-Diffusion), which consists of two main components: a compressing model and a novel mixed augmentation with fixed probability (MAFP) strategy. Specifically, the compressing model can constrain the complexity of the denoiser function hypothesis space and MAFP can effectively increase the training samples by providing more informative guidance than existing data augmentation methods in the compressed hypothesis space. Extensive experiments on several datasets demonstrate that LD-Diffusion can achieve better performance compared to other diffusion models. Codes are available at https://github.com/zzhang05/LD-Diffusion.

## 1. Introduction

Diffusion-based generative models (diffusion models) have decisively surpassed GANs in generative modeling, achieving superior image quality, greater diversity, and more stable training on image generation tasks (Dhariwal & Nichol, 2021; Ho et al., 2020; Karras et al., 2022). In theory, diffusion models aim to train a score-based model (Song et al., 2021) that matches the score function of the data distribution via denoising score matching (Song et al., 2021; Karras et al., 2022; Wang et al., 2023a). The current success of diffusion models in unconditional image synthesis (Vahdat et al., 2021; Song et al., 2023; 2020; Nichol & Dhariwal, 2021), text-to-image generation (Podell et al., 2023; Rombach et al., 2022a; Nichol et al., 2021; Saharia et al., 2022; Wang et al., 2023b) and audio generation (Kong et al., 2020; Popov et al., 2021) tasks is fueled by the almost unlimited supply of samples. Nevertheless, collecting and cleaning such data can be expensive, time-consuming, and sometimes infeasible, which significantly constrains the applications of diffusion models in real-world scenarios.

Focusing on training diffusion models with limited data, we present a novel theoretical insight for diffusion models that the denoising score matching error of all training samples, i.e., the total denoising score matching error, is determined by two key factors: the denoiser function hypothesis space and the number of training samples (§3.2). To further validate this theoretical insight, we conduct several experiments to illustrate that both factors significantly influence the training of diffusion models (§3.3). According to the unveiled insight, challenges arise when directly adapting existing methods (Karras et al., 2022; Song & Ermon, 2019; Ho et al., 2020; Dhariwal & Nichol, 2021; Karras et al., 2024) to limited data settings because they struggle to minimize the total denoising score matching error within their denoiser function hypothesis space.

To improve the training of diffusion models with limited data, we propose a new diffusion model called Limited Data Diffusion (LD-Diffusion). Motivated by the theoretical insight, LD-Diffusion consists of two main components. The first main component is a compressing model with the encoder and decoder obtained from different pre-trained VAE models, which aims to reduce the complexity of denoiser function hypothesis space. Although image compression models have previously been applied to several diffusion models (Esser et al., 2021; Vahdat et al., 2021; Rombach et al., 2022a), they all seek to accelerate the training speed and reduce the computational cost. In contrast, the com-

[1]Queen's University Belfast [2]Huawei UKRD. Correspondence to: Yang Hua <y.hua@qub.ac.uk>.

*Proceedings of the 42nd International Conference on Machine Learning*, Vancouver, Canada. PMLR 267, 2025. Copyright 2025 by the author(s).

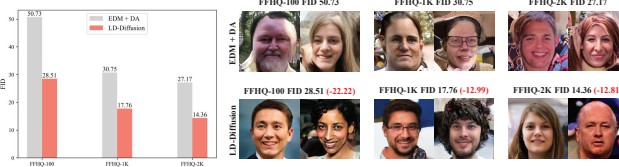

*Figure 1.* The comparison of EDM + DA[1] (Karras et al., 2022) and LD-Diffusion on the FFHQ (Karras et al., 2020b) dataset ($256 \times 256$) under limited data settings. Left: Compared Fréchet Inception Distance (FID) (Heusel et al., 2017) scores of LD-Diffusion and EDM + DA on the FFHQ dataset. The FID score of LD-Diffusion on FFHQ-100 is even lower than the FID score of EDM + DA on FFHQ-1K, which demonstrates that LD-Diffusion can achieve huge data efficiency compared with EDM + DA. Right: Comparison of images generated by EDM +DA and LD-Diffusion on the FFHQ dataset, highlighting the superior quality of images produced by the latter.

pressing model in LD-Diffusion aims to constrain the complexity of the denoiser function hypothesis space to obtain a low-dimensional hypothesis space that can better represent the diffusion-based high-dimensional hypothesis space (Wang et al., 2024; Boffi et al., 2024; Li et al.). Having such a compressed low-dimensional hypothesis space can make minimizing the total denoising score matching error easier with limited data (§4.1). The second main component in LD-Diffusion is a novel data augmentation called Mixed Augmentation with Fixed Probability (MAFP), which is designed for the compressed low-dimensional hypothesis space. Concretely, MAFP is more informative than existing data augmentation (DA) methods (Karras et al., 2022) in the compressed low-dimensional hypothesis space, and can effectively increase the number of training samples while avoiding leaking issues (Karras et al., 2020a; Zhang et al., 2020; Zhao et al., 2021; Zhang et al., 2024a). In addition, we introduce improved training techniques, i.e., patch training (Wang et al., 2023c) and Out-of-Distribution (OOD) regularization, to further boost performance under limited data settings. With these components and techniques, we select the most representative diffusion model, i.e., EDM (Karras et al., 2022), to implement LD-Diffusion. As illustrated in Figure 1, LD-Diffusion shows significant improvements compared with EDM + DA[1] (Karras et al., 2022) under limited data settings.

In summary, our paper is a pioneering study focusing on training diffusion models with limited data with three main contributions as follows: (1) We are the first to propose the novel theoretical insight for diffusion models that the total denoising score matching error is affected by two factors, i.e., denoiser function hypothesis space and the number of training samples. According to this theoretical insight, when training diffusion models with limited data, the total

---

[1]The EDM + DA is the method that EDM applies DA, where the DA (with a conditional input strategy) is proposed in EDM.

denoising score matching error is challenging to be minimized within the denoiser function hypothesis space of existing methods; (2) We propose a novel diffusion model called Limited Data Diffusion (LD-Diffusion), which consists of two main components and two improved training techniques. These components and techniques can effectively improve the training of diffusion models with limited data; (3) Experiments on FFHQ and low-shot (Zhao et al., 2020) datasets demonstrate that LD-Diffusion can achieve better performance compared to other diffusion models.

## 2. Preliminary

### 2.1. Diffusion-based Generative Models

In recent years, diffusion models (Karras et al., 2022; Ho et al., 2020; Dhariwal & Nichol, 2021) have exhibited their effectiveness in image-generation tasks. Suppose we are given a dataset $\{\boldsymbol{x}_n\}_{n=1}^{N}$, where each sample in the dataset is independently drawn from the data distribution $p_{data}(\boldsymbol{x})$. The goal of diffusion models is to construct a diffusion process $\{\boldsymbol{x}(t)\}_{t=0}^{T}$ indexed by a continuous time variable $t \in [0, T]$. According to Song et al. (2021), this diffusion process can be modeled as the solution to a stochastic differential equation (SDE):

$$d\boldsymbol{x} = \mathbf{f}(\boldsymbol{x}, t)dt + g(t)d\mathbf{w}, \qquad (1)$$

where $\mathbf{w}$ denotes a standard Brownian motion, $\mathbf{f}(\cdot, t) : \mathbb{R}^d \to \mathbb{R}^d$ is a vector-valued function called the drift coefficient of $\boldsymbol{x}(t)$, $g(t) \in \mathbb{R}$ is a real-valued function called the diffusion coefficient, and $dt$ represents a negative infinitesimal time step. Then, the reverse of a diffusion process is also a diffusion process, given by the reverse-time SDE, running backwards in time:

$$d\boldsymbol{x} = \left[\mathbf{f}(\boldsymbol{x}, t) - g(t)^2 \nabla_{\boldsymbol{x}} \log p_{\sigma_t}(\boldsymbol{x})\right] dt + g(t)d\overline{\mathbf{w}}, \quad (2)$$

where $\sigma_t$ is a schedule that defines the desired noise level at time $t$, $\overline{\mathbf{w}}$ is a standard Wiener process when time flows backwards from $T$ to $0$, and $d\overline{\mathbf{w}}$ can be viewed as infinitesimal white noise. The corresponding ordinary differential equation (ODE) of the reverse SDE is the probability flow ODE (Song et al., 2021), expressed as:

$$d\boldsymbol{x} = \left[\mathbf{f}(\boldsymbol{x}, t) - \frac{1}{2}g(t)^2 \nabla_{\boldsymbol{x}} \log p_{\sigma_t}(\boldsymbol{x})\right] dt, \qquad (3)$$

where $\nabla_{\boldsymbol{x}} \log p_{\sigma_t}(\boldsymbol{x})$ is the score function (Hyvärinen & Dayan, 2005), a vector field that points towards the higher density of data at a given noise level. The only unknown term in Eq.(2) and Eq.(3) is the score function. Thus, a function $\epsilon_\theta(\boldsymbol{x}, \sigma_t)$ parameterized by a learnable neural network, is applied to estimate the score function's values. Denoising score matching is currently the most popular way of estimating score functions applied in diffusion models. After learning the estimated score function $\epsilon_\theta(\boldsymbol{x}, \sigma_t)$, an estimated reverse SDE or ODE can be obtained in order to collect data samples from the estimated data distribution.

## 2.2. Diffusion Models with Limited Data

Recently, diffusion models have significantly outpaced GANs in generative modeling, delivering higher image quality, enhanced diversity, more stable training, and cutting-edge performance across various image generation tasks. However, training diffusion models commonly requires a large amount of data, which is expensive and difficult to clean. Although some studies (Wang et al., 2023c; Moon et al., 2022; Zhu et al., 2022; Hur et al., 2024; Wang et al., 2023a; Lu et al., 2023; Ruiz et al., 2023; Sinha et al., 2021b; Giannone et al., 2022; Yang et al., 2024) have attempted to finetune diffusion models with limited data, their approaches are similar to the transfer learning task and rely on the similarity of the source and target domain (Hur et al., 2024). In contrast, except for Patch Diffusion (Wang et al., 2023c), there are very few contributions that focus on training diffusion models with limited data from scratch. Specifically, Patch Diffusion introduces a diffusion model with the patch training strategy to reduce data and computational resources. However, experimental results show that Patch Diffusion still needs thousands of training samples. In this study, we aim to train robust diffusion models with only hundreds of samples from scratch, aligning with the training GANs with limited data (Zhang et al., 2025; 2024b).

## 2.3. Limited Data Settings in Generative Models

For limited data settings in GAN-based approaches (Zhao et al., 2020; Li et al., 2022), there are two common scenarios: **Case 1:** A small subset of a large dataset is used to train the model, while the full dataset serves as the reference distribution for calculating the FID. This case is applied to the experiments with the FFHQ dataset; **Case 2:** The original small dataset is used both for training and as the reference distribution for FID calculations. This case is typically utilized in experiments with low-shot datasets. In this study, we consider both cases when designing LD-Diffusion.

## 3. Theoretical Insight

### 3.1. Score-based Model and Denoising Score Matching

Without loss of generality, we choose the most representative diffusion model, i.e., EDM (Karras et al., 2022), for our analysis. Notably, EDM can be implemented using various diffusion architectures, such as noising conditional score network (NCSN) (Song & Ermon, 2019), denoising diffusion probabilistic model (DDPM) (Ho et al., 2020) and ablated diffusion model (ADM) (Dhariwal & Nichol, 2021). According to Song et al. (2021) and Karras et al. (2022), diffusion models aim to train a score-based model $\epsilon_\theta(\boldsymbol{x}, \sigma_t)$ that matches the score function of data distributions $\triangledown_{\boldsymbol{x}} \log p_{\sigma_t}(\boldsymbol{x})$. To achieve this goal, EDM builds up the denoiser function with preconditioning (Karras et al.,

2022), i.e., $D_\theta(\boldsymbol{x}, \sigma_t)$ parameterized by a neural network, to minimize the expected $L_2$ denoising error for samples drawn from the distribution $p_{data}$ independently for every $\sigma_t$. This can be expressed as:

$$\mathcal{L}(D_\theta; \sigma_t) = \mathbb{E}_{\boldsymbol{y} \sim p_{data}} \mathbb{E}_{\boldsymbol{n} \sim N(0, \sigma_t^2 I)} \|D_\theta(\boldsymbol{y} + \boldsymbol{n}; \sigma_t) - \boldsymbol{y}\|_2^2, \quad (4)$$

where $\boldsymbol{y}$ is a training image, $\boldsymbol{n}$ is noise and $\sigma_t$ is a schedule that defines the desired noise level. Then, the score-based model can be represented as:

$$\epsilon_\theta(\boldsymbol{x}, \sigma_t) = (D_\theta(\boldsymbol{x}, \sigma_t) - \boldsymbol{x})/\sigma_t^2. \quad (5)$$

Based on Appendix B.3 in EDM (Karras et al., 2022), by expanding the expectations in Eq.(4), it can be rewritten as an integral over the noisy sample $\boldsymbol{x}$:

$$\mathcal{L}(D_\theta; \sigma_t) = \int_{\mathbb{R}^d} \underbrace{\frac{1}{Y} \sum_{i=1}^{Y} \mathcal{N}\left(\boldsymbol{x}; \boldsymbol{y}_i, \sigma_t^2 \mathbf{I}\right) \|D_\theta(\boldsymbol{x}; \sigma_t) - \boldsymbol{y}_i\|_2^2}_{=:\mathcal{L}(D_\theta; \boldsymbol{x}, \sigma_t)} \, \mathrm{d}\boldsymbol{x}. \quad (6)$$

Eq.(6) demonstrates that we can minimize $\mathcal{L}(D_\theta; \sigma_t)$ by minimizing $\mathcal{L}(D_\theta; \boldsymbol{x}, \sigma_t)$ independently for each $\boldsymbol{x}$. Based on the theory presented in Song et al. (2021), Karras et al. (2022) and Wang et al. (2023c), with sufficient data provided, the optimal $D_\theta^*(\boldsymbol{x}; \sigma_t)$ can be expressed as:

$$D_\theta^*(\boldsymbol{x}; \sigma_t) = \arg \min_{D_\theta(\boldsymbol{x}; \sigma_t)} \mathcal{L}(D_\theta; \boldsymbol{x}, \sigma_t). \quad (7)$$

Based on Eq.(7), the optimal score-based model $\epsilon_\theta^*(\boldsymbol{x}, \sigma_t)$ in EDM can match the score function of data distributions $\triangledown_x \log p_{\sigma_t}(\boldsymbol{x})$ at any time $t$, which can be formulated as follows:

$$\epsilon_\theta^*(\boldsymbol{x}, \sigma_t) = (D_\theta^*(\boldsymbol{x}, \sigma_t) - \boldsymbol{x})/\sigma_t^2 = \triangledown_{\boldsymbol{x}} \log p_{\sigma_t}(\boldsymbol{x}). \quad (8)$$

### 3.2. Theoretical Analysis of Denoising Score Matching with Limited Data

Based on Eqs.(4), (6) and (7), we provide the theoretical analysis of denoising score matching with limited data. According to Wang et al. (2020), minimization of Eq.(6) can be analyzed by empirical risk minimization (Mohri et al., 2018; Vapnik, 1991) (with possible regularizers) in theory. For a better illustration, we define essential mathematical symbols as follows.

Let $D_\theta^*(\boldsymbol{x}, \sigma_t) = \arg \min_{D_\theta(\boldsymbol{x}; \sigma_t)} \mathcal{L}(D_\theta; \boldsymbol{x}, \sigma_t)$ be the function that minimizes the expected risk, $\hat{D}_\theta(\boldsymbol{x}, \sigma_t) = \arg \min_{D_\theta(\boldsymbol{x}; \sigma_t) \in \mathcal{D}} \mathcal{L}(D_\theta; \boldsymbol{x}, \sigma_t)$ be the function in $\mathcal{D}$ that minimizes the expected risk and $D_\theta^N(\boldsymbol{x}, \sigma_t) = \arg \min_{D_\theta^N(\boldsymbol{x}; \sigma_t) \in \mathcal{D}} \mathcal{L}(D_\theta^N; \boldsymbol{x}, \sigma_t)$ be the function in $\mathcal{D}$ that minimizes the empirical risk with $N$ samples provided. $\mathcal{D}$ is the denoiser function hypothesis space in practice and $N$ is the number of training samples.

As $D_\theta^*(\boldsymbol{x}, \sigma_t)$ is unknown, the common approach is to approximate it by certain $D_\theta(\boldsymbol{x}; \sigma_t) \in \mathcal{D}$. $\hat{D}_\theta(\boldsymbol{x}; \sigma_t)$ is the

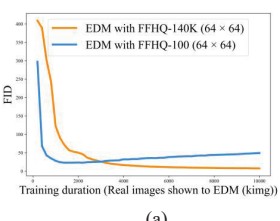 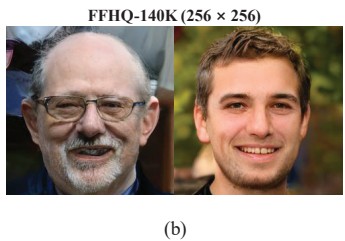 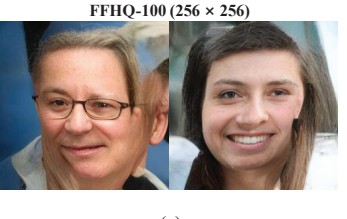

*Figure 2.* Validation experiments for the two factors highlighted in §3.2. Here, we select the EDM (Karras et al., 2022) with the FFHQ (Karras et al., 2020b) dataset for experiments: (a) The comparison of FID (Heusel et al., 2017) curves on EDM with the FFHQ-100 ($64 \times 64$) dataset and EDM with the FFHQ-140K ($64 \times 64$) dataset during training; (b) Generated images of EDM with FFHQ-140K ($256 \times 256$) dataset; (c) Generated images of EDM with the FFHQ-100 ($256 \times 256$) dataset; (d) Generated images of EDM with the FFHQ-100 ($64 \times 64$) dataset.

best approximation for $D_\theta(\boldsymbol{x}; \sigma_t) \in \mathcal{D}$, and $D_\theta^N(\boldsymbol{x}, \sigma_t)$ is the optimal hypothesis in $\mathcal{D}$ obtained by empirical risk minimization. For simplicity, we assume that $D_\theta^*(\boldsymbol{x}, \sigma_t)$, $\hat{D}_\theta(\boldsymbol{x}; \sigma_t)$ and $D_\theta^N(\boldsymbol{x}, \sigma_t)$ are unique. Then, the total error of $\mathcal{L}(D_\theta; \boldsymbol{x}, \sigma_t)$ in Eq.(6) during training can be formulated following existing approaches (Bottou & Bousquet, 2007; Bottou et al., 2018) as:

$$
\mathbb{E}\left[\mathcal{L}\left(D_\theta^N; \boldsymbol{x}, \sigma_t\right) - \mathcal{L}\left(D_\theta^*; \boldsymbol{x}, \sigma_t\right)\right] = \underbrace{\mathbb{E}\left[\mathcal{L}\left(\hat{D}_\theta; \boldsymbol{x}, \sigma_t\right) - \mathcal{L}\left(D_\theta^*; \boldsymbol{x}, \sigma_t\right)\right]}_{\mathcal{E}_{\text{app}}\,(\mathcal{D})}
$$
$$
+ \underbrace{\mathbb{E}\left[\mathcal{L}\left(D_\theta^N; \boldsymbol{x}, \sigma_t\right) - \mathcal{L}\left(\hat{D}_\theta; \boldsymbol{x}, \sigma_t\right)\right]}_{\mathcal{E}_{\text{est}}\,(\mathcal{D}, N)},
$$
(9)

where $\mathcal{E}_{\text{app}}(\mathcal{D})$ is the approximation error which measures how close the functions in $\mathcal{D}$ can approximate the optimal $D_\theta^*(\boldsymbol{x}, \sigma_t)$, and $\mathcal{E}_{\text{est}}(\mathcal{D}, N)$ is the estimation error which measures the effect of minimizing the empirical risk $\mathcal{L}(D_\theta^N; \boldsymbol{x}, \sigma_t)$ instead of the expected risk $\mathcal{L}(\hat{D}_\theta; \boldsymbol{x}, \sigma_t)$ in $\mathcal{D}$.

Eq.(9) demonstrates that the total error is affected by both $\mathcal{D}$ and $N$. In general, with sufficient $N$, $\mathcal{L}(D_\theta^N; \boldsymbol{x}, \sigma_t)$ can be a good approximation of $\mathcal{L}(D_\theta^N; \boldsymbol{x}, \sigma_t)$ with the empirical risk minimizer $D_\theta^N(\boldsymbol{x}, \sigma_t)$. Then, the $\mathcal{E}_{\text{est}}(\mathcal{D}, N)$ can be easily reduced in $\mathcal{D}$. However, with only limited $N$ provided in $\mathcal{D}$, the $\mathcal{L}(D_\theta^N; \boldsymbol{x}, \sigma_t)$ may then be far from being a good approximation of the expected risk $\mathcal{L}(\hat{D}_\theta; \boldsymbol{x}, \sigma_t)$. Thus, the empirical risk minimizer $D_\theta^N(\boldsymbol{x}, \sigma_t)$ is no longer reliable. In this case, based on Eq.(9), the total error of $\mathcal{L}(D_\theta; \boldsymbol{x}, \sigma_t)$ is difficult to minimize, which leads to a suboptimal score-based model that matches the score function of data distributions in practice, resulting in a decrease in the quality of generated images (shown in Figure 2 (c)).

### 3.3. Validation Experiments

To further demonstrate the theoretical insight in §3.2, we approach the experiments by artificially subsetting larger commonly-used datasets, i.e., FFHQ (Karras et al., 2020b),

following the studies of GANs with limited data (Karras et al., 2020a; Zhao et al., 2020). We select EDM as the baseline with FFHQ-100 and FFHQ-140K datasets for experiments. Specifically, FFHQ-100 is a subset of 100 images in the FFHQ dataset and FFHQ-140K is the full FFHQ dataset with xflip augmentation. The FFHQ dataset with 70K images is used as the reference distribution for the FID calculation. We measure quality by computing FID between 50K generated images and all available training images, as recommended by Heusel et al. (2017), regardless of the subset actually used for training. Following the EDM, the generated images are based on the same ODE solver, i.e., the number of function evaluations (NFE) is 79.

The influence of the $N$ on the training of the EDM is shown in Figure 2 (a), (b) and (c). Without loss of generality, we select the FFHQ dataset with different image resolutions (representing different dimensions of $\mathcal{D}$) for experiments. Figure 2 (a) compares the FID curves of EDM during training on the FFHQ-100 and FFHQ-140K datasets with $64 \times 64$ resolution. Figure 2 (b) and (c) compare the generated images of EDM with the FFHQ-140K ($256 \times 256$) and FFHQ-100 ($256 \times 256$) datasets. These results demonstrate that $N$ can significantly influence the quality of generated images with the same $\mathcal{D}$.

The effects of the $\mathcal{D}$ on the training of the EDM are shown in Figure 2 (c) and (d). Specifically, we select different image resolutions with the same $N$ for experiments. It is clear that the quality of generated images with EDM under the $256 \times 256$ FFHQ-100 dataset (Figure 2 (c)) is much worse than the quality of generated images with EDM under the $64 \times 64$ FFHQ-100 dataset (Figure 2 (d)), demonstrating that low-dimensional $\mathcal{D}$ is easier to optimize, leading to higher quality generated images.

## 4. Limited Data Diffusion (LD-Diffusion)

To address the challenges identified in our theoretical analysis in §3.2, we propose a novel diffusion model called

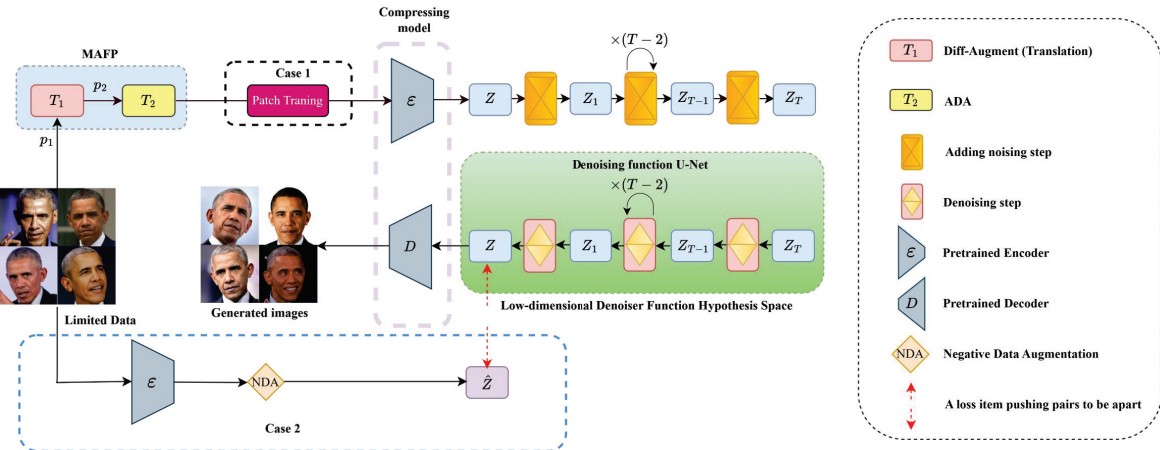

*Figure 3.* An overview of Limited Data Diffusion (LD-Diffusion). LD-Diffusion consists of two components: A compressing model with pre-trained encoder ($\varepsilon$) and decoder ($D$) models from similar but slightly different VAE models (with the same latent space) in Rombach et al. (2022b) and a novel data augmentation method called Mixed Augmentation with Fixed Probability (MAFP). The compressing model can obtain a compressed low-dimensional denoiser function hypothesis space by constraining its complexity. MAFP can effectively increase the number of training samples within the compressed low-dimensional hypothesis space. Furthermore, we apply patch training (Wang et al., 2023c) to Case 1 (§2.3) and propose novel Out-of-distribution (OOD) regularization for Case 2 (§2.3) to further improve the training of diffusion models with limited data.

Limited Data Diffusion (LD-Diffusion) to improve the training of diffusion models with limited data, as shown in Figure 3. LD-Diffusion has two components: (1) A compressing model that reduces the complexity of the $\mathcal{D}$, where $\mathcal{D}$ is the denoiser function hypothesis space in practice, same as in §3.2; (2) A novel data augmentation method, MAFP, designed to increase the number of training samples $N$ within the compressed low-dimensional $\mathcal{D}$. Furthermore, we introduce improved training techniques for LD-Diffusion with two different limited data cases (shown in §2.3). More theoretical interpretations of LD-Diffusion can be found in §A.1.

### 4.1. Compressing Model

The compressing model in LD-Diffusion is built using unpaired pre-trained encoder (Rombach et al., 2022d) and decoder (Rombach et al., 2022c) models to reduce the complexity of $\mathcal{D}$. The purpose of applying pre-trained encoder and decoder models is to constrain the complexity of $\mathcal{D}$, thereby resulting in a low-dimensional $\mathcal{D}$ that can better represent the diffusion-based high-dimensional hypothesis space (Wang et al., 2024; Boffi et al., 2024; Li et al.). According to Wang et al. (2020), the low-dimensional $\mathcal{D}$ consists of a small area to be considered for optimization. With the compressed low-dimensional $\mathcal{D}$, the limited $N$ is sufficient to reduce the total denoising score matching error (Germain et al., 2016; Mahadevan & Tadepalli, 1994; Nguyen & Zakynthinou, 2018). Notably, it has already been proven that the encoder and decoder models do not eliminate significant details (Rombach et al., 2022a). Thus, the compressing model can prevent the heavy leaking issue

(Karras et al., 2020a; Zhang et al., 2020; Zhao et al., 2021; Zhang et al., 2024a) during training. Furthermore, applying the pre-trained encoder and decoder models from the same pre-trained VAE models can unavoidably lead to a small loss of image details (Rombach et al., 2022a), which can limit the variety of detailed information in generated images, thereby leading to a decrease in performance. Therefore, we select the pre-trained encoder and decoder models from similar but slightly different VAE models (with the same latent space) for the compressing model in practice, which can further improve performance (See §B.2).

### 4.2. Mixed Augmentation with Fixed Probability

Although data augmentation (DA) methods have already been applied to EDM (Karras et al., 2022), this DA is designed based on ADA (Karras et al., 2020a) in the larger denoiser function hypothesis space. In contrast, following the proposed compressing model in §4.1, the DA for diffusion models with limited data should be more informative in the compressed low-dimensional $\mathcal{D}$. With the fact that Diff-Augment (Zhao et al., 2020) is more informative than ADA in the low-dimensional feature space (Sauer et al., 2021), combining Diff-Augment and ADA is the most straightforward approach to achieving this goal. However, directly combining Diff-Augment and ADA can cause a significant leaking issue. To address this, we propose a novel DA method called Mixed Augmentation with Fixed Probability (MAFP), which applies two fixed probabilities $p_1$ and $p_2$ with a conditional input strategy (Karras et al., 2022) to control Diff-Augment and ADA, respectively. Specifically, Diff-Augment (Translation) is applied with a probability $p_1$

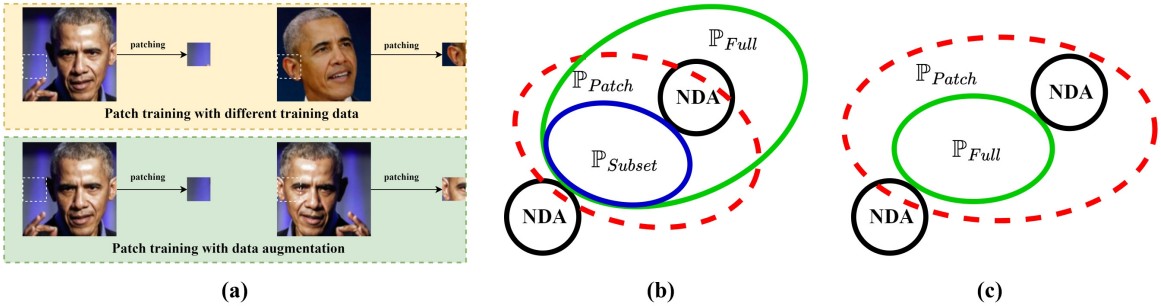

*Figure 4.* Understanding the improved training techniques, i.e., patch training and OOD regularization, on two scenarios. (a) Patch training leads to the leaking problem in two situations, i.e., different training data with patch training and augmentation data with patch training. Both situations result in different patches in patch-wise denoising score matching. (b) Schematic overview of the learning distributions with two techniques on Case 1 (§2.3). (c) Schematic overview of the learning distributions with two techniques on Case 2 (§2.3).

and skipped with a probability $1 - p_1$, and ADA is applied with a probability $p_2$ and skipped with a probability $1 - p_2$ in MAFP. As a result, MAFP can avoid unnecessary DA operations with a fixed probability during training, thereby effectively avoiding the leaking problem in the compressed low-dimensional denoiser function hypothesis space. Additionally, the proposed MAFP also has great generalization ability, allowing it to be applied to GAN-based methods to improve performance under limited data settings. More details of MAFP can be found in §A.2 and §B.3.

### 4.3. Improved Training Techniques for Two Cases of Limited Data Settings

Due to the difference in reference distribution for the two limited data cases (§2.3), the common leaking issue (Karras et al., 2020a; Zhang et al., 2020; Zhao et al., 2021; Zhang et al., 2024a) affects performance differently in each case. Specifically, for Case 1, leaking samples (Out-of-Distribution (OOD) samples) from the subset may actually be in-distribution for the full dataset, potentially improving training under limited data. Conversely, for Case 2, leaking samples are truly OOD samples for the dataset, which hinders the training under limited data. Based on the above analysis, we introduce the patch training (Wang et al., 2023c) for Case 1 and propose a novel OOD regularization for Case 2 to enhance the training of diffusion models with limited data, respectively.

**Patch training.** Recently, Patch Diffusion (Wang et al., 2023c) introduces patch training into diffusion models. Patch Diffusion presents the patch-wise denoising score matching to improve the training of diffusion models. However, patch training can unavoidably lead to leaking issues. Specifically, we observe that patch training in diffusion models can also result in different patches during patch-wise denoising score matching in two situations, as shown in Figure 4 (a). Both situations can lead to the selection of

different patches at the same image position during patch-wise denoising score matching. Sampling the images from these random different patches can lead to the leaking problem (Karras et al., 2020a; Zhang et al., 2020; Zhao et al., 2021; Zhang et al., 2024a), i.e., producing OOD samples, in the diffusion models.

**Out-of-Distribution (OOD) regularization.** Motivated by the success of the regularization method in GANs with limited data (Fang et al., 2022; Tseng et al., 2021), we propose a novel regularization method called OOD regularization. To the best of our knowledge, OOD regularization is the first regularisation method designed for diffusion models. Specifically, OOD regularization aims to penalize the gradient if the generated samples are close to the OOD samples. To achieve this, the novel denoiser score matching with OOD regularization is shown as follows:

$$\mathcal{L}(D_\theta; \sigma_t) = \mathbb{E}_{\boldsymbol{y} \sim p_{data}} \mathbb{E}_{\hat{\boldsymbol{y}} \sim \hat{p}_{data}} \mathbb{E}_{\boldsymbol{n} \sim N(0, \sigma_t^2 I)} [\underbrace{\|D_\theta(\boldsymbol{y} + \boldsymbol{n}; \sigma_t) - \boldsymbol{y}\|_2^2}_{\text{Original loss}}$$

$$+ \underbrace{\frac{1}{\|D_\theta(\boldsymbol{y} + \boldsymbol{n}; \sigma_t) - \hat{\boldsymbol{y}}\|_2^2 + \eta}}_{\text{Our OOD regularization}}],$$

(10)

where $\hat{\boldsymbol{y}}$ is the OOD samples which are obtained by applying the Negative Data Augmentation (NDA) (Sinha et al., 2021a) on the real samples $\boldsymbol{y}$, and $\eta$ is a constant 100 that aims to enable the loss function to become stable. Eq.(10) demonstrates that the proposed OOD regularization penalizes the gradient in denoiser score matching if generated samples are close to the OOD samples, which can effectively avoid the potential leaking issue.

**More explanation of applying patch training and OOD regularization in two cases.** To better understand how the leaking problem influences the training of diffusion models with limited data, the schematic overview of the learning distributions for patch training and OOD regularization in two limited data cases is shown in Figure 4 (b) and (c).

| Method | P? | FFHQ | | |
| --- | --- | --- | --- | --- |
| | | 100 | 1K | 2K |
| EDM (Karras et al., 2022) | No | 79.10 | - | - |
| EDM + DA (Karras et al., 2022) | No | 50.73 | 30.75 | 27.17 |
| Patch Diffusion (Wang et al., 2023c) | No | 44.45 | 28.03 | 25.32 |
| LPDM-8 (Wang et al., 2023c) | Yes | 32.78 | 19.67 | 15.47 |
| **LD-Diffusion (ours)** | Yes | **28.51** | **17.76** | **14.36** |

*Table 1.* FID scores (lower is better) of different methods on the $256 \times 256$ FFHQ dataset. Massive Augmentation (MA) (Cui et al., 2022), i.e., xflipping, is applied in all methods. The results of the EDM are directly from Adaptive IMLE (Aghabozorgi et al., 2023) while the results of EDM + DA are obtained based on the official open-source codes (Karras et al., 2022). The results of Patch Diffusion and LPDM-8 are obtained by ourselves based on official codes (Wang et al., 2023c). The downsampling factors in the pre-trained encoder and decoder are selected as 8 for LPDM-8 and LD-Diffusion. The FIDs are averaged over three runs; all standard deviations are less than 1%, relatively. Here, **P** represents the **Pre-training** model to facilitate the training of diffusion models in the latent space.

Figure 4 (b) demonstrates that the leaking issue caused by the patch-training can benefit the training of diffusion models when the reference distribution is based on the large-scale dataset, i.e., Case 1 (§2.3). Figure 4 (c) indicates that the leaking issue can decrease the training of the diffusion models when the reference distribution is based on small datasets, i.e., Case 2 (§2.3). Therefore, the patch training should be applied to Case 1 and the OOD regularization should be applied to Case 2.

# 5. Experiments

## 5.1. Datasets and Implementation Details

We select FFHQ (Karras et al., 2020b) and low-shot (Zhao et al., 2020) datasets for the experiments. For fair comparisons, we follow the official open-source codes[2] for pre-processing and resizing the FFHQ to $256 \times 256$, as used in existing studies (Karras et al., 2020a; Zhao et al., 2020; Li et al., 2022). We select the most representative diffusion model, i.e., EDM, as the baseline for LD-Diffusion. Specifically, the noising conditional score network (NCSN) (Song & Ermon, 2019) is selected as the architecture of the EDM baseline on low-shot datasets and the denoising diffusion probabilistic model (DDPM) (Ho et al., 2020) is selected as the architecture of the EDM baseline on the FFHQ dataset (refer to §5.4.4). For the compressing model in LD-Diffusion, the pre-trained encoder is selected from SD-MSE (Rombach et al., 2022d) and the pre-trained decoder is selected from SD-EMA (Rombach et al., 2022c) with a downsample factor of 8 (refer to §B.2). For the MAFP in LD-Diffusion, we set $p_1$ and $p_2$ as 0.1 for all experiments (refer to §5.4.2). Furthermore, patch training (Wang et al.,

---

[2] https://github.com/NVlabs/edm

2023c) is applied to LD-Diffusion on experiments with the FFHQ dataset. In contrast, the proposed OOD regularization is applied to LD-Diffusion in experiments with low-shot datasets. Based on EDM (Karras et al., 2022), the denoise sampling Number of Function Evaluations (NFE) is set as 79 and 511 for all diffusion-based methods on FFHQ and low-shot datasets, respectively.

## 5.2. Results on the FFHQ Dataset

We compare LD-Diffusion with other diffusion models with limited data by conducting experiments on the $256 \times 256$ FFHQ dataset. As shown in Table 1, LD-Diffusion achieves state-of-the-art (SOTA) performance compared to other diffusion models. To further demonstrate the effectiveness of LD-Diffusion, we also compare LD-Diffusion with other diffusion methods using Precision and Recall (Kynkäänniemi et al., 2019). As shown in Table 2, LD-Diffusion achieves SOTA Precision and Recall performance compared with other methods.

## 5.3. Results on Low-shot Datasets

We compare LD-Diffusion with all of the other state-of-the-art diffusion models by conducting experiments on the low-shot datasets. The results are shown in Table 3. LD-Diffusion outperforms other diffusion models and achieves SOTA performance. Additional comparisons of LD-Diffusion with other methods using Precision and Recall can be found in §B.6. To further demonstrate the superiority of LD-Diffusion, the comparisons of LD-Diffusion with other SOTA generative models with limited data can be found in §B.7. The compared generated images on low-shot datasets by EDM + DA and LD-Diffusion are shown in Figure 5.

## 5.4. Ablation Study

### 5.4.1. EFFECTIVENESS OF COMPONENTS IN LD-DIFFUSION

We conduct the ablation study to show the components of LD-Diffusion are effective in both limited data settings introduced in §2.3. The results are shown in Tables 4 and 5. It is clear that the design of LD-Diffusion (refer to Figure 3) is reasonable and can indeed improve performance.

### 5.4.2. EFFECTIVENESS OF FIXED PROBABILITY IN MAFP

To show the effectiveness of the fixed probability $p_1$ and $p_2$ in MAFP, we conduct an ablation study by selecting different values of probability $p_1$ and $p_2$ in LD-Diffusion. To achieve this goal, we first apply $p_2$ with ADA to LD-Diffusion in order to find the best value of $p_2$ (without $p_1$), and the results are shown in Table 6. Then, applying the best

| Method | FFHQ-100 | | FFHQ-1K | | FFHQ-2K | |
|---|---|---|---|---|---|---|
| | **P** | **R** | **P** | **R** | **P** | **R** |
| EDM + DA (Karras et al., 2022) | 0.724 | 0.005 | 0.668 | 0.222 | 0.709 | 0.281 |
| Patch Diffusion (Wang et al., 2023c) | 0.746 | 0.006 | 0.707 | 0.224 | 0.718 | 0.292 |
| LPDM-8 (Wang et al., 2023c) | 0.789 | 0.011 | 0.762 | 0.241 | 0.759 | 0.323 |
| **LD-Diffusion (ours)** | **0.791** | **0.018** | **0.767** | **0.249** | **0.769** | **0.323** |

*Table 2.* A comparison of Precision (**P**) and Recall (**R**) (Kynkäänniemi et al., 2019) (higher is better) of LD-Diffusion with other diffusion models on the $256 \times 256$ FFHQ dataset. Massive Augmentation (MA), i.e., xflipping, is applied to all methods. The **P** and **R** are averaged over three runs; all standard deviations are less than $1\%$, relatively.

| Method | Pre-training? | 100-shot | | | Animal-Face | |
|---|---|---|---|---|---|---|
| | | Obama | Grumpy | Panda | Cat | Dog |
| EDM (Karras et al., 2022) | No | 51.30 | 36.90 | 23.70 | 48.60 | 100.10 |
| EDM + DA (Karras et al., 2022) | No | 37.10 | 29.94 | 10.81 | 36.88 | 57.14 |
| Patch Diffusion (Wang et al., 2023c) | No | 41.47 | 30.89 | 13.25 | 43.71 | 72.17 |
| LPDM-8 (Wang et al., 2023c) | Yes | 14.27 | 14.56 | 5.13 | 14.92 | 15.95 |
| **LD-Diffusion (ours)** | Yes | **13.00** | **13.31** | **4.70** | **12.77** | **12.48** |

*Table 3.* FID scores (lower is better) of different methods on low-shot datasets ($256 \times 256$). We follow the setting used in (Zhao et al., 2020). Massive Augmentation (MA), i.e., xflipping, is applied in all methods. The FIDs are averaged over three runs; all standard deviations are less than $1\%$, relatively.

| Method | FFHQ-100 (Case 1) |
|---|---|
| LD-Diffusion | **28.51** |
| *w/* OOD regularization | 32.02 |
| LD-Diffusion | **28.51** |
| *w/o* Patch Training | 31.57 |
| *w/o* Compressing model | 49.29 |
| *w/o* MAFP | 79.10 |

*Table 4.* FID scores on the FFHQ-100 dataset ($256 \times 256$) by adding or gradually removing the corresponding component in LD-Diffusion. The FIDs are averaged over three runs; all standard deviations are less than $1\%$, relatively.

| Method | 100-shot Obama (Case2) |
|---|---|
| LD-Diffusion | **13.00** |
| *w/* Patch training | 13.97 |
| LD-Diffusion | **13.00** |
| *w/o* OOD regularization | 13.28 |
| *w/o* Compressing model | 36.31 |
| *w/o* MAFP | 51.30 |

*Table 5.* FID scores on the 100-shot Obama dataset ($256 \times 256$) by adding or gradually removing the corresponding component in LD-Diffusion. The FIDs are averaged over three runs; all standard deviations are less than $1\%$, relatively.

| $p_2$ | 0.0 | 0.1 | 0.2 | 0.4 | 0.8 |
|---|---|---|---|---|---|
| FID | 13.98 | **13.55** | 13.86 | 14.51 | 15.22 |

*Table 6.* FID scores on the 100-shot Obama dataset ($256 \times 256$) by selecting different $p_2$ with ADA. The FIDs are averaged over three runs; all standard deviations are less than $1\%$, relatively.

| $p_1$ | 0.0 | 0.1 | 0.2 | 0.4 | 0.8 |
|---|---|---|---|---|---|
| FID | 13.40 | **13.00** | 13.24 | 13.92 | 14.67 |

*Table 7.* FID scores on the 100-shot Obama dataset ($256 \times 256$) by selecting different $p_1$ with Diff-Augment (Translation) under best value $p_2 = 0.1$ with ADA. The FIDs are averaged over three runs; all standard deviations are less than $1\%$, relatively.

| Method | Obama | FFHQ-100 |
|---|---|---|
| LD-Diffusion (with DA in EDM) | 13.57 | 29.87 |
| LD-Diffusion (with **MAFP**) | **13.00** | **28.51** |

*Table 8.* FID scores on the 100-shot Obama and FFHQ-100 datasets ($256 \times 256$) for comparisons of MAFP with the existing DA in EDM using LD-Diffusion. The FIDs are averaged over three runs; all standard deviations are less than $1\%$, relatively.

value $p_2 = 0.1$ with ADA, we add $p_1$ with Diff-Augment (Translation) in LD-Diffusion to obtain the best value of $p_1$, and the results are shown in Table 7. Consequently, we set $p_1 = 0.1$ and $p_2 = 0.1$ for all experiments.

### 5.4.3. COMPARISON OF DIFFERENT DATA AUGMENTATIONS IN LD-DIFFUSION

We conduct experiments by applying MAFP and the existing DA to LD-Diffusion. The results are shown in Table 8, indicating that MAFP is more informative than the existing DA method (Karras et al., 2022) for LD-Diffusion.

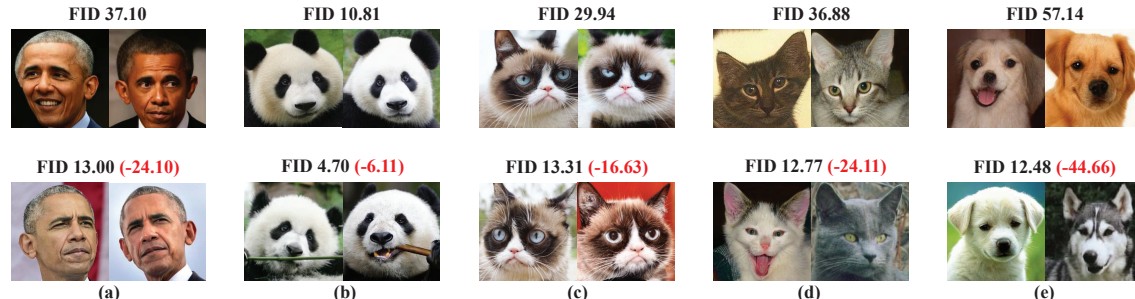

*Figure 5.* The comparison of generated images with LD-Diffusion and EDM + DA on (a) 100-shot Obama, (b) 100-shot Panda, (c) 100-shot Grumpy-cat, (d) AnimalFace-cat and (e) AnimalFace-dog datasets. **Top**: Images generated by EDM + DA. **Bottom**: Images generated by LD-Diffusion. The decreasing value of FID in red color demonstrates the improvement of LD-Diffusion over EDM + DA.

| Method | DDPM | ADM | NCSN |
|---|---|---|---|
| EDM + DA | **50.73** | 60.17 | 52.89 |
| LD-Diffusion | **28.51** | 33.39 | 29.59 |

*Table 9.* FID scores (lower is better) on the FFHQ-100 dataset with different selected architectures for EDM + DA and the proposed LD-Diffusion. The FIDs are averaged over three runs; all standard deviations are less than 1%, relatively.

| Method | DDPM | ADM | NCSN |
|---|---|---|---|
| EDM + DA | 38.09 | 45.89 | **37.10** |
| LD-Diffusion | 13.11 | 15.87 | **13.00** |

*Table 10.* FID scores (lower is better) on the 100-shot Obama dataset with different selected architectures for EDM + DA and the proposed LD-Diffusion. The FIDs are averaged over three runs; all standard deviations are less than 1%, relatively.

| Method | 100-shot Obama | FFHQ-100 |
|---|---|---|
| EDM (Karras et al., 2022) | 51.30 | 79.10 |
| EDM2 (Karras et al., 2024) | 112.26 | 119.72 |

*Table 11.* Compared FID scores (lower is better) of EDM and EDM2 on the 100-shot obama and FFHQ-100 dataset ($256 \times 256$).

### 5.4.4. COMPARISON OF DIFFERENT DIFFUSION ARCHITECTURES IN LD-DIFFUSION

To demonstrate how different diffusion model architectures influence the training of diffusion models with limited data, we conduct experiments by selecting different diffusion model architectures for EDM + DA and LD-Diffusion in two limited data cases. For Case 1, we study this by conducting experiments on the EDM + DA and LD-Diffusion with the FFHQ-100 dataset, and the results are shown in Table 9. DDPM (Ho et al., 2020) slightly outperforms NCSN (Song & Ermon, 2019) and achieves better performance compared to ADM (Dhariwal & Nichol, 2021). For Case 2, we approach this by conducting experiments on the EDM + DA and LD-Diffusion with the 100-shot Obama dataset. The results are shown in Table 10. NCSN (Song & Ermon, 2019) marginally outperforms DDPM (Ho et al., 2020) and demonstrates superior performance compared to ADM (Dhariwal & Nichol, 2021). The denoise sampling NFE is selected as 511 and 79 for the 100-shot Obama and FFHQ datasets, respectively. Furthermore, both Tables 9 and 10 show that

the proposed LD-Diffusion can significantly improve performance compared with EDM + DA with all three different diffusion architectures, demonstrating the generality of the proposed method. Additionally, one recent approach called EDM2 (Karras et al., 2024) has been proposed to scale up EDM (Implemented with the ADM architecture). To demonstrate that selecting EDM as the baseline in LD-Diffusion is reasonable, we conduct experiments comparing EDM and EDM2 on both limited data cases. The results are shown in Table 11. Given that EDM2 is implemented with the ADM architecture consisting of more parameters, it is reasonable that the results of EDM2 are worse than EDM in both limited data cases.

Additional ablation studies on utilizing various pre-trained encoders and decoders in the compressing model (§B.2), applying different augmentations in MAFP (§B.3), using diverse denoise sampling NFE during the sampling stage (§B.4) and applying different NDA methods in OOD regularization (§B.5) can be found in Appendix.

## 6. Conclusion

In this paper, we theoretically demonstrate that minimizing the total denoising score matching error is difficult to achieve under limited data within the denoiser function hypothesis space of existing methods. To address this, we further propose a new diffusion model called Limited Data Diffusion (LD-Diffusion) to improve the training of diffusion models with limited data. Extensive experiments on several datasets demonstrate the superiority of LD-Diffusion.

## Acknowledgement

This work is partially supported by the High Performance Computing Centre of Queen's University Belfast and the Kelvin-2 supercomputer (EPSRC grant EP/T022175/1). The first author is supported by a scholarship of the Queen's University Belfast and the Department for the Economy, Northern Ireland.

## Impact Statement

This paper is a pioneering study focusing on training diffusion-based generative models with limited data from scratch. The technical contributions of this paper do not raise any particular ethical challenges. However, because technology is usually a double-edged sword, our work may also bring potential social risks when applying diffusion-based generative models with limited data. For example, it may ease the high-quality fake media synthesis using only limited data.

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

# A. More Explanation of LD-Diffusion

## A.1. Theoretical Interpretations for LD-Diffusion

**ODE formulation of LD-Diffusion.** Based on the existing studies (Wang et al., 2023c; Sinha et al., 2021a), the proposed improved training techniques do not influence the diffusion model learning data distributions, demonstrating that the proposed improved training techniques will not influence the convergence of diffusion models. Then, following the ODE formulation in Song et al. (2021), Karras et al. (2022) and Vahdat et al. (2021), the probability flow ordinary differential equation (ODE) of LD-Diffusion can be formulated as:

$$\mathrm{d}\boldsymbol{x} = -\dot{\sigma}_t \sigma_t \nabla_{\varepsilon(T(\boldsymbol{x}))} \log p_{\sigma_t}(\varepsilon(T(\boldsymbol{x}))\mathrm{d}t, \tag{11}$$

where $T$ is proposed MAFP, $\sigma_t$ is a schedule that defines the desired noise level at time $t$ and $\varepsilon$ is the pre-trained encoder as in Rombach et al. (2022d). According to §4, both $\varepsilon$ and $T$ would not result in the leaking issue.

**Denoising score matching for LD-Diffusion.** According to the denoising score matching in Karras et al. (2022), we build up the denoiser function, i.e., $D_\theta(\boldsymbol{x}; \sigma_t)$, that minimizes the expected $L_2$ denoising error for samples drawn from distribution $p(\varepsilon(T(data)))$ independently for every $\sigma_t$. Following the two limited data cases in §2.3, the denoising score matching for Case 1 is based on the patch diffusion (Wang et al., 2023c) and can be expressed as:

$$\mathcal{L}(D_\theta; \sigma_t) = \mathbb{E}_{\boldsymbol{y} \sim p(\varepsilon(T(data)))} \mathbb{E}_{\boldsymbol{n} \sim N(0, \sigma_t^2 I)} \mathbb{E}_{(i,j,s) \sim \mu}$$
$$\left\| D_\theta(\boldsymbol{y}_{i,j,s} + \boldsymbol{n}; \sigma_t) - \boldsymbol{y}_{i,j,s} \right\|_2^2, \tag{12}$$

where $\boldsymbol{n}$ is noise, $\varepsilon$ is the pre-trained encoder model, $T$ is the proposed MAFP, $\mu$ denotes the uniform distribution on the corresponding value range and $\boldsymbol{y}_{i,j,s}$ is the randomly crop small patches for any $\boldsymbol{y} \sim p(\varepsilon(T(data)))$. Specifically, $(i, j)$ is left-upper corner pixel coordinates to locate each image patch, and $s$ denotes the patch size. Then, following Eq.(12), the denoising score matching for Case 2 can be expressed as:

$$\mathcal{L}(D_\theta; \sigma_t) = \mathbb{E}_{\boldsymbol{y} \sim p(\varepsilon(T(data)))} \mathbb{E}_{\hat{\boldsymbol{y}} \sim \hat{p}_{\varepsilon(data)}} \mathbb{E}_{\boldsymbol{n} \sim N(0, \sigma_t^2 I)}$$
$$[\underbrace{\| D_\theta(\boldsymbol{y} + \boldsymbol{n}; \sigma_t) - \boldsymbol{y} \|_2^2}_{\text{Original loss}} + \underbrace{\frac{1}{\| D_\theta(\boldsymbol{y} + \boldsymbol{n}; \sigma_t) - \hat{\boldsymbol{y}} \|_2^2 + \eta}}_{\text{Our OOD regularization}}], \tag{13}$$

where $\boldsymbol{y}$ is a training image and $\hat{\boldsymbol{y}}$ is the OOD sample that is obtained by applying the NDA on the real samples $\boldsymbol{y}$, and $\eta$ is a constant 100 that aims to enable the loss function to become stable. Eq.(12) and Eq.(13) are the denoising score matching for FFHQ and low-shot datasets, respectively. Both Eq.(12) and Eq.(13) demonstrate that the denois-

ing score matching for LD-Diffusion is in the compressed low-dimensional denoiser function hypothesis space, which consists of a smaller area to be considered for optimization, thus resulting in better performance with limited data provided.

According to Wang et al. (2023c) and Sinha et al. (2021a), the patch training and proposed OOD regularization do not influence the learning data distribution, demonstrating that our proposed improved training techniques will not influence the convergence of the score-based model. Then, following Karras et al. (2022) and Vahdat et al. (2021), there still exists an optimal $D_\theta^*(\varepsilon(T(\boldsymbol{x})); \sigma_t)$ for Eq.(12) and Eq.(13) in theory. In this case, based on Eq.(5) and Eq.(8), the optimal score-based model $\epsilon_\theta^*(\varepsilon(T(\boldsymbol{x})), \sigma_t)$ can match the score function $\nabla_{\varepsilon(T(\boldsymbol{x}))} \log p_{\sigma_t}(\varepsilon(T(\boldsymbol{x})))$ at any time $t$, which can be expressed as:

$$\epsilon_\theta(\varepsilon(T(\boldsymbol{x})), \sigma_t) = (D_\theta^*(\varepsilon(T(\boldsymbol{x})); \sigma_t) - \varepsilon(T(\boldsymbol{x})))/\sigma_t^2$$
$$= \nabla_{\varepsilon(T(\boldsymbol{x}))} \log p(\varepsilon(T(\boldsymbol{x})); \sigma_t), \tag{14}$$

where $\sigma_t$ is a schedule that defines the desired noise level. Eq.(14) demonstrates that the score-based model can also reach an optimal in the compressed low-dimensional denoiser function hypothesis space in theory.

## A.2. Generalization of Proposed Mixed Augmentation with Fixed Probability (MAFP)

Based on §4.2, to demonstrate the generalization of the proposed MAFP, we apply it to Generative Adversarial Networks (GANs), following the existing studies (Karras et al., 2020a; Jiang et al., 2021). Specifically, we set the probabilities $p_1$ and $p_2$ in MAFP to be controlled adaptively by the overfitting degree of $D$ (Karras et al., 2020a; Jiang et al., 2021), denoted as MAFP (Adaptive). The results are presented in Table 12, revealing that MAFP (Adaptive) outperforms Diff-Augment (Zhao et al., 2020) and ADA (Karras et al., 2020a) on low-shot datasets with StyleGAN2 (Karras et al., 2020b).

# B. More Experiment Results

## B.1. Experimental Implementation Details

We conduct all the experiments on a single workstation with two A5000 (24G) GPUs, with a total of 10 same workstations for all experiments. We follow the EDM[3] to build up our software environment. We set the overall training duration for all diffusion-based generative models on the FFHQ dataset and low-shot datasets as 20000kimgs and 40000kimgs, respectively. To reduce the inference time for the diffusion models, we evaluate the model per

---

[3]https://github.com/NVlabs/edm

| Method | MA | 100-shot | | | Animal-Face | |
|---|---|---|---|---|---|---|
| | | Obama | Grumpy Cat | Panda | Cat | Dog |
| StyleGAN2 + Diff-Augment (Zhao et al., 2020) | Yes | 46.87 | 27.08 | 12.06 | 42.44 | 58.85 |
| StyleGAN2 + ADA (Karras et al., 2020a) | Yes | 45.69 | 26.62 | 12.90 | 40.77 | 56.83 |
| StyleGAN2 **+ MAFP (Adaptive)** | Yes | **41.13** | **25.87** | **10.93** | **38.69** | **54.15** |

*Table 12.* FID score (lower is better) on several low-shot datasets ($256 \times 256$). We follow the setting as in Zhao et al. (2020). Massive Augmentation (Cui et al., 2022) is applied to all of the methods. The FIDs are averaged over three runs; all standard deviations are less than 1%, relatively.

| Compressing Model | Obama | AF-Dog |
|---|---|---|
| Both encoder and decoder from SD-EMA (Rombach et al., 2022c) | 13.31 | 12.85 |
| Encoder from SD-EMA and decoder from SD-MSE (Rombach et al., 2022d) | 13.20 | 12.71 |
| Both encoder and decoder from SD-MSE | 13.13 | 12.63 |
| Encoder from SD-MSE and decoder from SD-EMA | **13.00** | **12.48** |

*Table 13.* FID score (lower is better) by selecting different pre-trained encoders and decoders in the compressing model in LD-Diffusion. The FIDs are averaged over three runs; all standard deviations are less than 1%, relatively.

500kimgs for the FFHQ dataset and per 1000kimgs for low-shot datasets. The snapshot with the best FID for each method is reported in the experiments.

### B.2. Impact of Different Pre-trained Encoder and Decoder in Compressing Model

To demonstrate which pre-trained encoder and decoder are suitable for the compressing model in LD-Diffusion, we conduct experiments by selecting different pre-trained encoders and decoders with downsample factors 8 in the compressing model, and the results are shown in Table 13. It is clear that the pre-trained encoder and decoder in SDE-MSE and SD-EMA, respectively, can lead to the best performance. Additionally, we do not select the pre-trained encoder and decoder in SDXL (Podell et al., 2023) because they can only work on full precision, i.e., FP32, rather than half precision, i.e., FP16, which can significantly increase computational resources for all of the experiments.

### B.3. More Experiments on MAFP

According to §4.2, the ablation study of Diff-Augment (Zhao et al., 2020) in MAFP on the 100-shot Obama dataset is shown in Table 14. It is clear that Diff-Augment (Translation) achieves the best performance compared with other settings. Therefore, we apply the Diff-Augment (Translation) in MAFP for LD-Diffusion.

### B.4. More Ablation Studies of Denoising Sampling Number of Score Function Evaluations (NFE)

We conduct experiments by varying the denoising sampling NFE during the inference stage for LD-Diffusion. The results are presented in Table 16. The selection of NFE aligns with the different denoising sampling NFE settings in EDM (Karras et al., 2022). It is evident from the results that setting NFE=79 for the FFHQ dataset and NFE=511 for low-shot datasets in the main paper is reasonable.

### B.5. More Ablation Studies of NDA in OOD Regularization

We conduct an ablation study by selecting different NDA methods, i.e., Jigsaw, Stitching, Mixup (Zhang et al., 2018), Cutmix (Yun et al., 2019) and Cutout (DeVries & Taylor, 2017) as in NDA-GAN (Sinha et al., 2021a), for the proposed OOD regularization in LD-Diffusion. The results of the 100-shot Obama dataset are shown in Table 17. Jigsaw achieves better performance compared with other NDA methods in the proposed OOD regularization.

### B.6. More Experimental Results on Low-shot Datasets

According to §5.3, we present the comparative results of LD-Diffusion and other methods using Precision and Recall on low-shot datasets, as shown in Table 15. LD-Diffusion outperforms other diffusion models and always achieves the best Precision compared with other diffusion models. This strongly demonstrates that the generated image distribution produced by LD-Diffusion has a high likelihood of falling into the real data distribution, indicating the superiority of LD-Diffusion. Moreover, Patch Diffusion achieves the highest Recall score in most cases. This is attributed to the fact that Patch Diffusion applies the patch training and does not utilize pre-trained models to compress difficult information within image distributions. Consequently, Patch Diffusion endeavors to learn such difficult information with various patches, increasing the likelihood of the real distribution falling into the generated image distribution compared to

| Method | FID |
|---|---|
| No Diff-Augment in MAFP for LD-Diffusion | 13.32 |
| Diff-Augment (Translation) in MAFP for LD-Diffusion | **13.00** |
| Diff-Augment (Color) in MAFP for LD-Diffusion | 13.36 |
| Diff-Augment (Cutout) in MAFP for LD-Diffusion | 16.15 |
| Diff-Augment (Translation + Color) in MAFP for LD-Diffusion | 13.11 |
| Diff-Augment (Translation + Color + Cutout) in MAFP for LD-Diffusion | 15.01 |

*Table 14.* FID score (lower is better) on the 100-shot Obama ($256 \times 256$) by selecting different Diff-Augment cases in the MAFP module in LD-Diffusion. Massive Augmentation (Cui et al., 2022) is applied to all of the settings. The FIDs are averaged over three runs; all standard deviations are less than 1%, relatively.

| Method | Obama | | Grumpy Cat | | Panda | | AnimalFace-Cat | | AnimalFace-Dog | |
|---|---|---|---|---|---|---|---|---|---|---|
| | **P** | **R** | **P** | **R** | **P** | **R** | **P** | **R** | **P** | **R** |
| EDM + DA | 0.965 | 0.380 | 0.870 | 0.330 | 0.874 | 0.360 | 0.969 | 0.306 | 0.955 | 0.188 |
| Patch Diffusion | 0.952 | **0.410** | 0.861 | **0.370** | 0.866 | **0.392** | 0.961 | **0.324** | 0.943 | 0.201 |
| LPDM-8 | 0.991 | 0.180 | 0.990 | 0.100 | 0.998 | 0.120 | 0.997 | 0.146 | 0.997 | 0.772 |
| LD-Diffusion (ours) | **0.996** | 0.100 | **0.994** | 0.080 | **1.000** | 0.012 | **0.999** | 0.103 | **1.000** | **0.773** |

*Table 15.* A comparison of Precision (**P**) and Recall (**R**) of LD-Diffusion with other diffusion models on the $256 \times 256$ low-shot datasets. Massive Augmentation (MA) is applied to all of the methods. The Precision and Recall are averaged over three runs; all standard deviations are less than 1%, relatively.

| Number of NFE | FID (100-shot Obama) | FID (FFHQ-100) |
|---|---|---|
| NFE=35 | 13.56 | 32.61 |
| NFE=79 | 13.32 | **28.51** |
| NFE=511 | **13.00** | 30.11 |

*Table 16.* FID score (lower is better) on the 100-shot Obama dataset ($256 \times 256$) and FFHQ-100 dataset ($256 \times 256$) by applying different numbers of denoising sampling NFE during the inference stage in LD-Diffusion. The FIDs are averaged over three runs; all standard deviations are less than 1%, relatively.

| NDA method in OOD regularization | FID |
|---|---|
| Jigsaw | **13.00** |
| Stitching | 13.24 |
| Mixup (Zhang et al., 2018) | 14.12 |
| Cutmix (Yun et al., 2019) | 13.47 |
| Cutout (DeVries & Taylor, 2017) | 13.29 |

*Table 17.* FID score (lower is better) on the 100-shot Obama dataset ($256 \times 256$) by selecting different NDA methods for OOD regularization in LD-Diffusion. Massive Augmentation (Cui et al., 2022) is applied to all of the settings. The FIDs are averaged over three runs; all standard deviations are less than 1%, relatively.

diffusion-based generative models with pre-trained models without patch training, thereby resulting in a higher Recall score.

## B.7. Comparisons of LD-Diffusion with Other Generative Models

To further demonstrate the superiority of our proposed LD-Diffusion, we compare it with other state-of-the-art (SOTA) generative models with limited data, namely Diffusion-Projected GAN (Wang et al., 2023d) and RS-IMLE (Vashist et al., 2024). The results, presented in Table 18, show that LD-Diffusion achieves performance comparable to other SOTA generative models. Notably, LD-Diffusion attains the best FID scores on the AnimalFace-Cat and AnimalFace-Dog datasets.

## B.8. Computational Cost

The comparison of the training and inference time of EDM + DA and LD-Diffusion on the 100-shot Obama dataset have been demonstrated in Table 19. LD-Diffusion can reduce the huge training and inference time compared with EDM + DA.

| Method | 100-shot | | | Animal-Face | |
|---|---|---|---|---|---|
| | Obama | Grumpy | Panda | Cat | Dog |
| Diffusion Projected GAN (Wang et al., 2023d) | **10.54** | 15.13 | **3.39** | 17.86 | 17.22 |
| RS-IMLE (Vashist et al., 2024) | 14.00 | **11.50** | 3.50 | 15.90 | 23.10 |
| **LD-Diffusion (ours)** | 13.00 | 13.31 | 4.70 | **12.77** | **12.48** |

*Table 18.* Compared FID scores (lower is better) of LD-Diffusion with other state-of-the-art (SOTA) generative models on low-shot datasets ($256 \times 256$). We follow the setting used in Zhao et al. (2020). Massive Augmentation (MA), i.e., xflipping, is applied in all methods. The FIDs are averaged over three runs; all standard deviations are less than 1%, relatively.

| Method | Training by 50K images (min) | Generating 5K images during inference (min) |
|---|---|---|
| EDM + DA | 40 | 165 |
| **LD-Diffusion (ours)** | **7** | **28** |

*Table 19.* The training and inference time of EDM + DA and LD-Diffusion on the 100-shot Obama dataset ($256 \times 256$). The half precision, i.e., FP16, is applied to all methods. The results are calculated by averaging over ten times on the two NVIDIA A5000 GPUs. All standard deviations are less than 1%, relatively.

## C. Discussion

### C.1. Discussion about Pre-trained Models and Limited Data Settings

This paper proposes a compressing model with the pre-trained encoder and decoder models in Rombach et al. (2022b) to constrain the complexity of the denoiser function hypothesis space, thus improving the training of the diffusion model with limited data. Some viewpoints may think that applying pre-trained models could break the limited data rules because pre-trained models are trained on a large dataset. We argue this from two aspects. First, applying pre-trained models to generative models to improve their training under limited data has been widely used in GANs. For example, the well-known approaches Projected GAN (Sauer et al., 2021) and vision-aided GAN (Kumari et al., 2022) have both applied the pre-trained models to improve the training with limited data. Second, our goal aims to train diffusion models with limited data from scratch. The pre-trained models applied to the LD-Diffusion are based on the pre-trained VAE models, which are not related to the diffusion models. Therefore, the proposed LD-Diffusion is trained with limited data from scratch and does not break the limited data rules.

### C.2. Discussion of Ethical Issues

This paper applies the 100-shot Obama dataset, i.e., the dataset consists of Obama faces, in the experiment section. This dataset is widely and commonly used without limitations in Data Effcient GANs (DE-GANs) research, and we follow the recent existing DE-GANs research to apply them to diffusion models with limited data. Furthermore, a lot of recent studies (Zhao et al., 2020; Cui et al., 2022; Chen et al., 2021; Li et al., 2022; Zhang et al., 2024a) on DE-GANs have applied this dataset in their experiments, demonstrating its application is reasonable and does not raise any ethical issues.

### C.3. Limitation

According to §B.1, to reduce the computational cost during the inference stage of diffusion models, we evaluate the model per 500kimgs for the FFHQ dataset and per 1000kimgs for low-shot datasets. The snapshots with the best FID for each method are reported in the experiments. This evaluation strategy is fair and can indeed reduce the computational cost. However, it can also make the reported results in the paper slightly worse than the best result.

