# OpenReview forum: "Training Diffusion-based Generative Models with Limited Data"
_ICML.cc/2025/Conference — ICML 2025 poster_

### Official Review · Reviewer_txiC · 2025-02-22

**Overall Recommendation:** 3

**Summary:**

The paper identifies that many existing diffusion models depend on large corpus of training image, and attempt to offer solutions to enable high quality training with only a fraction of the dataset size. The authors reveal the different cases for how a diffusion model can be trained with limited data, and construct their solution around two main components: latent compression and mixed augmentation with fixed probability.

**Claims And Evidence:**

- The claims made in the submission are supported.
 - The paper's method section could be expanded on expense of the rest of the paper. Specifically, I found the core of the algorithm in secs. 4.1 and 4.2, and 4.3 are short and hard to follow, and crucial details are deferred to the appendix. The main manuscript should be self-contained.

**Essential References Not Discussed:**

I cannot think of any.

**Experimental Designs Or Analyses:**

- The experiments seem sound to me.
- The results shown in this work are encouraging.

**Methods And Evaluation Criteria:**

Yes, the method and evaluation make sense for the problem.

**Other Comments Or Suggestions:**

No other comments.

**Other Strengths And Weaknesses:**

Strengths:
 - The limited fata regime is a key problem with diffusion models, as many distributions contain limited examples. Thus managing to reach exisiting sampling quality with much fewer samples can have high impact on the field.
 - The paper is comprehensive, offering modified solutions depending on the framing of the problem

**Questions For Authors:**

- Could the authora shed light on their choice to use 511 NFEs for generation with the small datasets? Is this crucial for high quality generation? How do the results change if the original 79 steps are used instead?

**Relation To Broader Scientific Literature:**

- As far as I understood, the selection of vae models out of the ones trained in previous works is done using a small ablation. This should be clearly explained in the main paper. Also, as using latent diffusion is standard practice, I do not see this as contribution of this work.

**Theoretical Claims:**

- The OOD regularization is presented without sufficient theoretical justification. The diffusion training is connected to the noisy distribution score, chaning the loss in this fashion may break some of these properties.

---

> ### Author Rebuttal · Authors · 2025-04-01
>
> **Q1. The paper's method section could be expanded on expense of the rest of the paper.**
>
> Due to the 8-page limitation of ICML submission, we have included additional details of our proposed LD-Diffusion in the Appendix. We will extend the Sections. 4.1, 4.2, and 4.3 based on the Appendix to provide more details of our proposed LD-Diffusion in the revised version (9 pages).
>
> **Q2. The OOD regularization is presented without sufficient theoretical justification.**
>
> The OOD regularization is one of the improved training techniques in our paper, which is mainly based on empirical experiments to further avoid the leaking issue during training. Specifically, OOD regularization aims to maintain the core properties of the diffusion models while introducing the necessary adjustments to enhance performance under limited data settings. We have conducted complete experiments to demonstrate that the changes to the loss function caused by OOD regularization do not significantly affect the stability or other important properties of the diffusion models. Additionally, as highlighted in the results and ablation studies, the LD-Diffusion demonstrates further improvement with OOD regularization on low-shot datasets.
>
> **Q3. The selection of VAE should be clearly explained in the main paper.**
>
> We have already explained the reason for selecting the pre-trained encoder and decoder models from similar but slightly different VAE models (with the same
> latent space) for the compressing model in Lines 264-274 (left). Furthermore, we conduct an ablation study for this selection in  Appendix B.2. We will add more details from the Appendix B.2 to the main paper in the revised version.
>
> **Q4. I do not see applying latent diffusion model as contribution of this work.**
>
> Although the latent diffusion model has been applied in some existing approaches, these studies primarily focus on accelerating training speed and reducing computational cost. In contrast, our work is the first to apply the latent diffusion model based on theoretical analysis (refer to Sec 3.2) to improve the training of diffusion models with limited data. We believe this presents a novel perspective distinct from existing methods and offers valuable insights for future research.
> Furthermore, based on Reviewer aWCW comments: The evaluation of applying these techniques (including latent diffusion model) to train diffusion models in limited data settings has never been done. This reinforces the viewpoint that applying the latent diffusion model to enhance diffusion model training with limited data can be regarded as a meaningful contribution in our paper.
>
> **Q5. The choice of NFE in the LD-Diffusion.**
>
> We have already provided a detailed ablation study of how to choose the NFEs for two cases in Table 16 in the Appendix. Specifically, the selection of NFE aligns with the denoising sampling NFE settings in EDM. Although the results in Table 16 show that different NFE selections do not significantly impact performance and are therefore not crucial for high-quality generation, it is evident from the results that setting NFE=79 for the FFHQ dataset and NFE=511 for low-shot datasets is reasonable and achieves the best practical. As shown in Table 16, if the NFE is selected as 79 for the low-shot datasets, it can decrease the performance accordingly.

---

### Official Review · Reviewer_aWCW · 2025-03-10

**Overall Recommendation:** 3

**Summary:**

This paper proposes a method to train diffusion models under settings with limited training data.

The suggested method includes several modifications to the standard diffusion model, including:
1. Learning diffusion in an embedding space (that comes from a pre-trained VAE model),
2. Applying penalty for producing bad samples (OOD regularization), and
3. Applying a combination of data augmentation techniques in the embedding space.

Each of these components exists already, but the specific application of the combination of components to training diffusion models with limited-data is novel, and the problem is of importance.

The method is evaluated against a baseline diffusion model and shown to perform better based on various metrics, including FID score.

Several ablation studies are also presented to systematically evaluate the different components of the proposed method.

## Update after rebuttal
Keeping same rating. Partially satisfied with some answers, not satisfied with answer to Q5 which I feel is important.

**Claims And Evidence:**

Claim 1:
The authors claim to be providing a novel theoretical insight into training with limited samples “We are the first to propose the novel theoretical insight for diffusion models that the total denoising score matching error is affected by two factors, i.e., denoiser function hypothesis space and the number of training samples”.

I find this insight to not be novel. This is already a well established insight from the paper referred to by the authors. The authors are simply interpreting the statement in their context - this is also not novel because the insight relates to learning of any function using empirical risk minimization, which is trivially applicable to almost any neural network method including this one.

Claim 2:
The proposed method differs from related works that use transfer learning for diffusion in limited-data settings.

I believe this claim needs to be substantiated a lot more. The basic premise of the claim is this: works that use transfer learning to finetune diffusion models assume that there is a diffusion model that works in a domain that is not very different from the target limited-data domain. The presented work trains models directly from the limited data. However, the authors are using a pre-trained VAE model to first convert the data to an embedding space. There may be an implicit assumption about the behaviour of the VAE model - which depends on some similarity between the dataset that the VAE was trained on, and the new dataset to learn. If this claim is to be substantiated, it must be shown that there are no requirements on the VAE - i.e. it is trained on a very different domain.

If this claim cannot be substantiated, then the method must also be compared with the transfer learning based methods.

Claim 3:
The proposed method improves over the selected baseline diffusion model in low-data settings.

This claim is mostly reasonable and is supported by a variety of experiments demonstrating it.

**Essential References Not Discussed:**

N/A

**Experimental Designs Or Analyses:**

The experimental design is very good, except for the exclusion of transfer learning baselines.

**Methods And Evaluation Criteria:**

Methods:
The specific changes suggested over the baseline diffusion model are all reasonable and well motivated.

Evaluation criteria:
The benchmark datasets and metrics used are sensible and sufficiently comprehensive.

**Other Comments Or Suggestions:**

1. There is one issue regarding the evaluation of the FFHQ-100 dataset. The evaluation labeled as “Case 1” suffers with the OOD regularization. Does that suggest that OOD regularization cannot be applied as is everywhere? Is that something that needs to be tuned?
2. Is there any intuition behind using different VAE encoder and decoders?

**Other Strengths And Weaknesses:**

Strengths:
1. The experimental section is very comprehensive (for the specific baseline method used). It covers tests on several datasets.
2. The ablation studies are really nice and help give valuable insights.

**Questions For Authors:**

1. What is novel about the theoretical insight given over the already published insight that is referred to in the paper?
2. How is the performance affected if the VAE model is trained on a very different domain (not natural images, e.g.)?
3. How does the performance compare with the transfer learning methods, if the initial diffusion model is trained on the same dataset that the VAE is trained on?

**Relation To Broader Scientific Literature:**

This paper presents a combination of several existing components in training of diffusion models. All of the components have been studied in semi-related contexts:
1. Training diffusion models over embedding spaces
2. The augmentation techniques Diff-Augment and NDA
3. OOD regularization

However, the novelty lies in the following:
1.Regularization has been attempted with GANs, not with diffusion models. Therefore, the empirical behaviour of regularization is not well understood.
2. The evaluation of these techniques to train diffusion models in limited data settings has never been done.

While I have questions about the assumptions regarding the VAE and comparisons with transfer learning related methods, the paper does present a recipe that works under the stated assumptions. It is certainly useful.

**Theoretical Claims:**

The theoretical insight presented by the authors is not novel, it is already established.

---

> ### Author Rebuttal · Authors · 2025-04-01
>
> **Q1. One issue of the FFHQ-100 dataset.**
>
> The purpose of designing OOD regularization is to mitigate the potential leaking issue in diffusion models under limited data conditions, thus OOD regularization is deliberately designed and can only be applied to the scenarios where
>  aiming to prevent the leaking issue. As illustrated in Sec 2.3, the evaluation of two limited data cases differs, which causes the leaking of augmentation issue to play a different role in the performance of each case (shown in Lines 278-291 and
> Appendix A.1). Under this situation, OOD regularization should be applied to low-shot datasets (Case 1) to mitigate the leaking problem, whereas it should not be applied to the FFHQ dataset (Case 2). Therefore, no tuning is needed for the OOD regularization when using it.
>
> **Q2. Is there any intuition behind using different VAE encoder and decoders?**
>
> We have already explained the intuition behind using slightly different encoder and decoder models (with the same latent space) in Lines 264-274. Specifically, using the same pre-trained VAE encoder and decoder can lead to a small loss of image details, which limits the variety of detailed information in generated images thus decreasing performance. By applying slightly different encoder and decoder models (with the same latent space), we intuitively alleviate this issue by providing more varied information without affecting convergence. The experiments presented in Appendix B.2 further validate this intuition.
>
> **Q3. What is novel about the theoretical insight?**
>
> Although empirical risk minimization is a well-established theory in machine learning, it has not been fully explored in diffusion models. Our paper extends the empirical risk minimization and provides pioneering theoretical insight into diffusion models with limited data. Notably, the proposed theoretical insight provides a different viewpoint compared with the commonly overfitting viewpoint in other generative models (such as GANs and IMLEs) with limited data. Therefore, we believe the proposed theoretical insight can inform future studies on diffusion models and even other generative models with limited data.
>
> **Q4. How is the performance affected if the VAE model is trained on a very different domain (not natural images, e.g.)?**
>
> The primary purpose of the pretrained VAE model in LD-Diffusion is to project images from the pixel space into a latent space while ensuring effective reconstruction back to the image domain. To achieve this, VAE models are typically trained on large-scale datasets such as LAION-Aesthetics, which provide diverse and high-quality natural images for the training of VAE models. Due to there is lack of such large-scale datasets from other domains for training the VAE model, we conduct experiments comparing EDM + DA and LD-Diffusion with non-natural image domain limited data, i.e., the BRECAHAD dataset containing 162 real-world medical images, to enable the pretrained VAE data and limited data are in the different domains. We follow the official codes of EDM for preprocessing and resizing the BRECAHAD data into 3240 partially overlapping crops with resolution $256 \times 256$ for avoiding the huge computational cost in the experiments.
>
> The results clearly illustrate that the pretrained VAE model applied in LD-Diffusion can still achieve significant improvements on non-natural image domain limited data such as medical image datasets. This demonstrates that the effectiveness of LD-Diffusion does not depend on domain similarity, which is different from transfer learning methods.
>
>  **Method**         | **FID (BRECAHAD)** |
> |---------------|---------------|
> | EDM + DA      | 45.49         |
> | LD-Diffusion  | **12.21**         |
>
> **Table 1:** Comparison of LD-Diffusion and EDM + DA using FID scores on the BRECAHAD dataset.
>
> **Q5. Comparing with the transfer learning methods?**
>
> According to our response in Q4, we argue that directly comparing the proposed LD-Diffusion (a training-from-scratch method) to transfer learning approaches is not a relevant evaluation. Below, we outline three key reasons supporting this perspective: (1) LD-Diffusion is a diffusion model trained from scratch (the parameters in Pretrained VAE models are frozen during training), differing from diffusion models that rely on transfer learning for fine-tuning with limited data. (2) As stated in Q4, the effectiveness of LD-Diffusion is not dependent on the similarity between the pretrained VAE model data and the applied limited data. In contrast, diffusion models that rely on transfer learning for fine-tuning typically require the similarity between the source data domain and the target limited-data domain to achieve improvements. (3) To the best of our knowledge, there are no open-source pretrained unconditional diffusion models that have been trained on the same dataset as the pretrained VAE, such as LAION-Aesthetics. Thus, it is difficult to build up a fair comparison baseline.

---

### Official Review · Reviewer_eEHg · 2025-03-16

**Overall Recommendation:** 3

**Summary:**

This paper focuses on the challenges of training diffusion-based models with minimal data. The authors introduce a novel model called Limited Data Diffusion (LD-Diffusion), which includes a compressing model to reduce hypothesis space complexity and a new data augmentation strategy called Mixed Augmentation with Fixed Probability (MAFP). This approach significantly improves the training and performance of diffusion models on limited data sets by effectively managing the hypothesis space and enhancing data utility. Extensive experiments across several data sets showcase LD-Diffusion's superior performance compared to existing methods.

**Claims And Evidence:**

Yes, the claims made in the submission are supported by detailed experimental evidence.

**Essential References Not Discussed:**

The related work section provides a comprehensive review of relevant literature, offering valuable context for understanding the study.

**Experimental Designs Or Analyses:**

How sensitive is the model performance to the choice of hyperparameters, especially those related to the compressing model and MAFP?

Were any hyperparameter optimization techniques employed, and how were the optimal parameters selected?

**Methods And Evaluation Criteria:**

The proposed method appears reasonable, with evaluation based on FID scores and Improved Precision and Recall metrics, which are appropriate for the problem. However, it would have been beneficial if the authors had also compared baseline approaches using the Inception score.

**Other Comments Or Suggestions:**

The paper is well-written but a few editorial stuff could be improved for better readability. For example, Equations 1, 2, and 3 are hard to read. Texts under Section 3.1 seem odd, i.e. gap between words is not evenly spaced, etc.

**Other Strengths And Weaknesses:**

**Strengths:**
Robustness through Design Choices: The model's architecture, integrating elements like patch training and OOD regularization tailored for different scenarios (Cases 1 and 2 as defined in the study), shows a deep technical understanding of how to enhance robustness and reliability in generative modeling (that’s what I felt at least).

Flexible Architecture: The flexibility of LD-Diffusion to be implemented with various underlying architectures (such as DDPM, ADM, and NCSN) illustrates its adaptability. This flexibility is a substantial technical strength as it allows the model to be tailored to the specific needs and constraints of different applications and datasets.

Theoretical Insights: The work provides novel theoretical insights into the impact of hypothesis space and training sample size on the denoising score matching error, contributing to a deeper understanding of the underlying dynamics of diffusion models.


**Weaknesses:**
Scalability: The proposed method, while effective for limited data scenarios, might not scale efficiently when the amount of data increases substantially. The techniques that compress the hypothesis space could potentially restrict the model's learning capacity in data-rich environments.

Complexity and Overhead of MAFP: While the MAFP strategy enhances training data utility, it also adds complexity and computational overhead to the training process. The need to manage and tune fixed probabilities for different augmentation techniques might complicate the training process, especially for users without extensive technical expertise.

Generalization Concerns: The compressing model's approach to reducing the hypothesis space complexity could theoretically limit the expressiveness of the generative model. By constraining the complexity, there's a risk that the model might not capture the full variability of data, particularly for complex or highly varied datasets, potentially affecting the generalizability of the model.

Impact of Data Augmentation on Authenticity: The introduction of synthetic data through MAFP could potentially lead to the generation of less authentic samples. The balance between enhancing data utility and maintaining the authenticity and quality of generated outputs is not extensively discussed, which might be critical for applications where fidelity is paramount.

**Questions For Authors:**

Could they provide more details on the computational resources required for training and deploying LD-Diffusion?

How does the model perform in terms of training time and resource utilization compared to traditional diffusion models?

Were any control experiments conducted to compare the performance of LD-Diffusion without the use of pre-trained components or the MAFP technique? This would help in understanding the individual contributions of these components to the overall model performance.

**Relation To Broader Scientific Literature:**

By addressing the practical challenge of training with limited data, the paper targets a significant barrier in the deployment of generative models, making it relevant for many applications in industry and academia.

**Theoretical Claims:**

The authors do not make any theoretical claims.

---

> ### Author Rebuttal · Authors · 2025-04-01
>
> **Q1. How sensitive of hyperparameters related to the compressing model and MAFP?**
>
> For the hyperparameters in the compressing model, we follow the default settings from the pretrained VAE models in Stable Diffusion. Regarding MAFP, we conduct experiments using fixed probabilities $p_1$ and $p_2$, with the results shown in Tables 6 and 7. It is evident that different choices of $p_1$ and $p_2$ can influence performance, but the impact is not significant. Specifically, increasing the optimal $p_1$ and $p_2$ values 2 $\times$ only affects 2% relative performance. For comparison, in Table 6 of the APA [1] paper, increasing the threshold hyperparameter
> $t$ by a factor of 1.33 results in a 7% relative change in performance.
>
> [1] Deceive d: Adaptive pseudo augmentation for gan training with limited data. NeurIPS 2021.
>
> **Q2. How were the optimal parameters selected?**
>
> Based on our response in Q1, aside from default hyperparameters, LD-Diffusion only uses $p_1$ and $p_2$ in the MAFP, with optimal values determined through experiments (Tables 6 and 7).
>
>
> **Q3. Scalability.**
>
> Our paper primarily focuses on research in limited data scenarios. In this case, LD-Diffusion employs the compressing model to constrain the hypothesis space, which can significantly enhance performance.
> For data-rich settings, Stable Diffusion has already shown that constraining the hypothesis space can significantly accelerate the training speed with minor and acceptable performance drops, demonstrating the scalability of compressing the hypothesis space.
>
> **Q4. Complexity and Overhead of MAFP.**
>
> Existing methods have already shown that both Diff-Augment and ADA introduce only negligible complexity and computational overhead in GANs. MAFP combines Diff-Augment and ADA with a fixed probability also incurs only a negligible increase in complexity and computational cost. Furthermore, MAFP is a plug-in module, non-technical can easily follow the default augmentation methods and fixed probability values provided in our paper to enhance performance under limited data settings.
>
> **Q5. Generalization Concerns.**
>
> Following our response in Q3, our paper primarily focuses on research in limited data scenarios. In Secs 2.2 and 2.3, we have shown that minimizing the total denoising score matching error in diffusion models is challenging under limited data conditions. In such cases, the high complexity and variability of the limited data further exacerbate the difficulty for diffusion models in learning meaningful information. To address this issue, LD-Diffusion employs a compressing model to constrain the hypothesis space complexity, leading to significant improvements under limited data settings. The substantial performance gains observed in our experiments demonstrate that the benefits of reducing hypothesis space complexity far outweigh its potential limitations. The insight of constraining the hypothesis space complexity can be also used to understand the effectiveness of other generative models under limited data, such as Projected GAN, further demonstrating the generalization ability of our approach.
>
> **Q6. Impact of Data Augmentation on Authenticity.**
>
> The problem you mentioned is the well-known leaking of augmentation issue, i.e., augmentation samples falling out of distribution, which leads to the generation of less authentic samples. In Lines 87 and 268-269 (right side), we have already stated that the design of MAFP aims to increase training samples while avoiding this issue. Additionally, improved training techniques such as OOD regularization can further mitigate this issue.
>
> **Q7. Computational resources required for LD-Diffusion.**
>
> Regarding the computational resources required for training and deploying LD-Diffusion, a single GPU with 24GB of memory is sufficient. This is because LD-Diffusion is benefiting from the use of the proposed compressing model.
>
> **Q8. How does the model perform in terms of training time and resource utilization compared to traditional diffusion models?**
>
> The comparison of training and sampling time between LD-Diffusion and EDM + DA can be found in Table 20. In terms of computational resources, benefiting from the use of the proposed compressing model, LD-Diffusion reduces GPU memory usage by nearly 8$\times$ under the same batch size and decreases GFLOPs by approximately 6$\times$ compared with EDM + DA.
>
> **Q9. Were any control experiments?**
>
> We have already presented ablation studies to demonstrate the effectiveness of each component in LD-Diffusion in Tables 4 and 5. By progressively removing each component (i.e., improved training techniques, compressing model and MAFP), the results help illustrate the individual contributions of the proposed components to the overall model performance, enhancing the understanding of their importance.
>
> **Due to the 5,000-character limit, The results of Inception Scores can be found in https://anonymous.4open.science/r/Inception-Scores/README.md.**

---

### Official Review · Reviewer_ivbF · 2025-03-23

**Overall Recommendation:** 4

**Summary:**

The authors propose a method of training diffusion models with limited data. The proposed method uses a compressing model to constrain the complexity of the denoiser function hypothesis space and a mixed augmentation with a fixed probability (MAPF) strategy to provide more informative guidance.

**Claims And Evidence:**

Yes

**Essential References Not Discussed:**

N/A

**Experimental Designs Or Analyses:**

Yes. I checked the experiments on FFHQ and low-shot datasets.

**Methods And Evaluation Criteria:**

Yes

**Other Comments Or Suggestions:**

N/A

**Other Strengths And Weaknesses:**

Strengths:
1. The authors propose a new method LD-Diffusion for training diffusion models on limited data which achieves better performance compared to other diffusion models.
2. The manuscript is well-written and easy to follow.
3. The authors conduct comprehensive experiments and ablation studies on FFHQ and low-shot datasets, demonstrating the effectiveness of the proposed compressive module and mixed augmentation with fixed probability (MAFP) module.
4. The experiments of the compressing model (Figure 2 c-d and Table 4-5) show the fact that training the diffusion model on limited data would be more informative in the compressed low-dimensional denoiser function hypothesis space.

Weakness:
1. Figure 4 shows how the leaking issues are different for case 1 and case 2. I think it might be helpful in understanding the proposed LD-Diffusion to explain how these differences in leaking issues lead to different strategies for cases 1 & 2, while I also think Figure 4 is critical for the motivation of the proposed method and needs to be in the main body.

**Questions For Authors:**

N/A

**Relation To Broader Scientific Literature:**

N/A

**Theoretical Claims:**

Yes. I checked equations 4~10 and didn't find any obvious issues.

---

> ### Author Rebuttal · Authors · 2025-04-01
>
> **Q1. Figure 4 shows how the leaking issues are different for case 1 and case 2. I think it might be helpful in understanding the proposed LD-Diffusion to explain how these differences in leaking issues lead to different strategies for cases 1 & 2, while I also think Figure 4 is critical for the motivation of the proposed method and needs to be in the main body.**
>
> Thank you for your valuable suggestions. We have already analyzed that the leaking issue affects performance differently for Cases 1 and 2 in Lines 278-291. Therefore,  we introduce the patch training for Case 1
> and propose a novel Out-of-distribution (OOD) regularization for Case 2 to enhance the training of diffusion models with limited data, respectively. Furthermore, due to space constraints, Sec A.1 and Figure 4 are currently included in the Appendix. In the revised version, we will reallocate space to incorporate them into the main paper, improving clarity and accessibility.

---

### Decision · Program_Chairs · 2025-05-01

**Decision:**

Accept (poster)

**Comment:**

This paper addresses an important problem of training diffusion models with limited data. The authors first show in theory and also in practice how the denoising score matching loss is affected by two factors, including the latent space and the number of training samples. With such insights, the authors propose compressing to a low-dimensional space and a mixed augmentation scheme.

While latent diffusion model is not new, this paper investigates latent space from the perspective of data-efficient diffusion models. Reviewers appreciate that the method can be applied to EDM and other variants of diffusion models. Experiments are convincing with large margin compared to previous methods. This paper also shows how to apply the method to two different settings.

Reviewer aWCW raised concerns about the lack of comparison to transfer learning. The setting of training from scratch is more challenging than transfer learning, i.e., fine-tuning. The authors mentioned the practical difficulty of training an unconditional diffusion model on the data that VAE is trained on.

This paper has advanced the state of the art of diffusion model with limited data. However, the framework seems to be a mix of variants of existing components such as compressing model, Diff-Augment and ADA augmentation, and patch training. The AC recommends to accpe this paper, but the authors should highlight their contributions in the final version.